# Integrated Transcriptomics Profiling in Chahua and Digao Chickens’ Breast for Assessment Molecular Mechanism of Meat Quality Traits

**DOI:** 10.3390/genes14010095

**Published:** 2022-12-28

**Authors:** Mohammed Abdulwahid Alsoufi, Yong Liu, Changwei Cao, Jinbo Zhao, Jiajia Kang, Mengyuan Li, Kun Wang, Yang He, Changrong Ge

**Affiliations:** 1College of Animal Science and Technology, Yunnan Agricultural University, Kunming 650201, China; 2Department of Animal Production, Faculty of Agriculture, Sana’a University, Alwehdah Street, Sana’a P.O. Box 19509, Yemen; 3Department of Food Science and Engineering, College of Biological Sciences, Southwest Forestry University, Kunming 650224, China

**Keywords:** Chahua chicken, Digao broiler chicken, breast muscle, meat quality traits, pathways and genes, transcriptomics

## Abstract

Meat quality traits are an important economic trait and remain a major argument, from the producer to the consumer. However, there are a few candidate genes and pathways of chicken meat quality traits that were reported for chicken molecular breeding. The purpose of the present study is to identify the candidate genes and pathways associated with meat quality underlying variations in meat quality. Hence, transcriptome profiles of breast tissue in commercial Digao (DG, 5 male) and Chahua (CH, 5 male) native chicken breeds were analyzed at the age of 100 days. The results found 3525 differentially expressed genes (DEGs) in CH compared to DG with adjusted *p*-values of ≤0.05 and log2FC ≥ 0.1 FDR ≤ 0.05. Functional analysis of GO showed that the DEGs are mainly involved in the two types of processes of meat quality, such as positive regulation of the metabolic process, extracellular structure organization, collagen trimer, cellular amino acid metabolic process, cellular amino acid catabolic process, and heme binding. Functional analysis of KEGG showed that the DEGs are mainly involved in the two types of processes of meat quality, such as oxidative phosphorylation, carbon metabolism, valine, leucine, and isoleucine degradation, and fatty acid degradation. Many of the DEGs are well known to be related to meat quality, such as *COL28A1, COL1A2, MB, HBAD, HBA1, ACACA, ACADL, ACSL1, ATP8A1, CAV1, FADS2, FASN, DCN, CHCHD10, AGXT2, ALDH3A2,* and *MORN4*. Therefore, the current study detected multiple pathways and genes that could be involved in the control of the meat quality traits of chickens. These findings should be used as an essential resource to improve the accuracy of selection for meat traits in chickens using marker-assisted selection based on differentially expressed genes.

## 1. Introduction

China is the largest meat market in the world, approximately 18% of chicken production in the world is consumed in China. Likewise, today’s projection research has forecasted that the enlargement of the poultry market is expected to continue in the future. Furthermore, consumers prefer chicken meat, particularly breast meat, because of its delicious taste, healthy nutritional content, ease of preparation, and ability to meet modern consumer needs [1,2]. Broiler chickens have been selected over the last few decades to improve carcass traits and increase thigh and breast yields. As a result, meat quality traits have been negatively impacted, and some undesirable meat quality traits have emerged, such as a decrease in taste quality and sensory adequacy for consumers. Meat quality is a complete concept that includes tenderness, flavor, nutrition, appearance, and other factors. However, many factors influence meat quality, including breed, feeding, management system, sex, and age, with the breed being the most important [3]. The CH Chicken breed, a native chicken breed in China, including the Yunnan chicken breed, has a unique meat quality compared to the DG broiler chicken and is extremely popular in Yunnan due to its meat characteristics. For this reason, the explanation of the molecular basis of meat quality in CH and DG chicken breeds will have an economic and biological effect. Furthermore, there are multiple genes that regulate meat quality traits in chicken, and some genes have been reported as associated with this trait, such as *TYRP1, TYR, DCT* [4], *AGA, COL28A1, COL1A2* [5], *MB, HBAD, HBA1* [6]*, ACACA, ACADL, ACSL1, ATP8A1, CAV1, FADS2, FASN* [7], DCN, *CHCHD10* [8], and *MORN4* [9].

In earlier decades, the main methods for identifying genes associated with meat quality and carcass trait performance in chickens were genome-wide association studies, quantitative trait locus mapping, and candidate gene analysis. As a result, many genomic regions and genetic associations have been identified. Similarly, genome-wide association studies can be used to identify the variants and genes that control economic traits [10,11] as they have been for meat quality [5,12] and carcass traits [13,14,15]. Although these methods have contributed significantly to our improved understanding of the underlying mechanisms of genetic traits, some potential constraints are still present. The important restriction is a good mapping needed to identify relevant variations. Furthermore, some biological pathways or novel genes linked with the targeted trait can be excluded accidentally. Next-generation sequencing (NGS) technologies have been widely used in recent years to identify differential expression as well as potential novel transcripts [16]. Based on the preceding discussion, RNA-Seq technology has been widely used to identify differentially expressed genes between two gene expression patterns, alternative splicing events, and causative variants.

Transcriptomic approach analysis is a foundation tool for gene function and structure studies [17], and it has become a fairly standard methodology in genetic studies, with growing applications in animal breeding [18]. General transcriptomes or complete transcripts include messenger RNA, ribosomal RNA, and transfer RNA [19,20]. RNA-sequencing analysis (transcriptional sequencing) has rapidly developed over the past few years, a technology of deep sequencing for transcriptome analysis [21]. This technology can detect the whole transcriptome at a single nucleotide level. Transcript expression and structure can provide analysis by RNA-seq technology. Moreover, the approach to gene expression could contribute to the systematic studies undertaken by evaluating the suitability of pathways representing significantly responsible genes responsible for genetic characteristics [22,23].

Recently, many studies have reported using RNA-seq technology [15,24] or microarray technology [25] on chicken transcriptomes. Besides that, chicken transcriptome analysis has already been performed for fatty acid metabolism [8], ultimate pH (pHu) [9], growth [26,27], feed efficiency [28], meat quality and muscle growth [29]. Wu et al. [30] performed RNA-seq technology of leg muscle between fast and slow-growing chickens of Jinghai yellow chickens with different body weights at 300 days to study the molecular mechanism of chicken growth. Between fast and slow-growing chickens, 87 genes were found to be differentially expressed. Cui et al. [31] used microarray technology to identify differentially expressed genes related to intramuscular fat (IMF) metabolism in Beijing-you chicken breast and thigh at 42 and 90 days of age, and 515 DEGs and 36 DEGs related to IMF metabolism were identified between the breast and thigh at 42 and 90 days. Furthermore, molecular pathways and genes related to regulating thigh meat quality and feed efficiency of Korat chickens were examined using microarray technology. The functional analysis emphasized numerous enriched function pathways, such as metabolic processes, immune responses, biological processes, and nucleotide metabolism [25]. The study explored biological genes related to controlling melanogenesis pathways in the breast muscles of Wu Liangshan black-boned chickens and broiler chickens. The findings identified 25 differentially expressed genes related to the melanogenesis pathways [4]. Zhang et al. [27] showed that 4608 differentially expressed genes (DEGs) were obtained by comparison of the Jinghai yellow using RNA-seq to study the transcriptome of the breast muscle. Karimi et al. [28] identified target genes that relate to feed efficiency by analyzing transcriptome profiles of liver tissue in native and commercial chicken breeds (Ross vs. native breeds) using RNA-Seq data. However, most of these studies have been focused on the broilers and layers of chicken breeds, still, the understanding of the mechanisms underlying meat quality and muscle development due to Chinese chicken breeds has remained limited. Chahua (CH2) is a new line of chickens. It was created through five generations of crossbreeding and selection from the CH Chicken breed and other breeds in Yunnan Province. It was established to develop the CH chicken breed to promote farming, ensure food safety in small villages, and maintain local chicken breeds. Slow-growing CH chickens are commonly slaughtered on the I2 to 16 weeks of age. Its meat is recognized as high quality [32,33]. However, the slow growth of CH chickens causes high production costs [34].

On the other hand, Digao chicken is a commercial chicken breed (broiler). Yunling Guangda Breeding Poultry Feed Company imported this breed from Australia in 1997 (Yunling Guangda Breeding Poultry Feed Co., Ltd., Kunming, China). At present, DG chicken breeds are the only products in Yunnan province. This breed is characterized by a variety of production, strong adaptability, fast growth rate, and delicious meat, it is suitable for breeding and market needs at all levels. The parent breeders of this breed are red feathers, with a unique “recessive white feather” genetic trait, which can be used as the mother of local breed improvement. The body weight of a 56-day-old commercial chicken with rapid feather growth and easy identification of males and females (male for 3–3.2 kg, female 2.8–3 kg). For this reason, the improvement of CH chickens genetically to increase growth rate is required with maintaining high meat quality for this breed.

The perspectives for a genetic enhancement between local and commercial chicken breeds are completely different. Furthermore, local chickens are destined for high-quality chicken meat; for this reason, these breeds are characterized by high meat quality. As study subjects, we used 100-day-old DG and CH chicken breeds. The Chahua chicken breed and the Digao chicken breed are two distinct breeds, with the Chahua chicken having high meat quality and the Digao chicken having low meat quality when compared to the Chahua chicken. The aim of the research was to determine the pathways and genes that may be involved in the meat quality formation of breast muscle of these two breeds.

## 2. Materials and Methods

### 2.1. Experimental Chickens and Sample Collecting

A total of 60 chickens each (one day old) of CH chickens (from an experimental farm, College of Animal Science and Technology, Yunnan Agricultural University, Kunming, China) and DG chickens (a commercial broiler line, from a Chicken Farm of Yunling Guangda Breeding Poultry Feed Co., Ltd., Kunming, China) were used. All of the chickens were placed in the experimental unit at the Yunnan Agricultural University’s experiment station. The birds were reared during two periods, the first period ranging from 1 day to 30 days, the chickens were under standard conditions with feed and water provided ad libitum and a starter diet (12.8 MJ/kg of ME, 20.6% CP). In the second period, the chickens were fed a regular diet (12.5 MJ/kg of ME, 18.4% CP) and had free water access for a period of 30 to 100 days (Appendix A). The diets of chicken were designed to satisfy the nutrient requirements of chicken [35] and recommendations of Chinese chicken feeding standers. The chickens were vaccinated for diseases, the temperature was 35 °C to 2 days of age and decreased gradually to 22 °C in 30 days, and humidity was maintained at 45%. Lighting was controlled by fluorescent lighting with a light:dark cycle of 12 h each.

At a sampling age of 100 days, 60 chickens from each breed were transported to the slaughtered unit at Yunnan agricultural university’s laboratory in the college of animal science and technology. The water and feed were withdrawn at 12, and 16 h, respectively, and chickens’ body weight was determined before the slaughter process. The slaughter process was conducted according to the requirements of the National Experimental Animal Slaughter Standard of China by the cervical dislocation method. In total, 60 samples of the pectoralis major breast muscle (200 mg) from each breed were harvested and put in an RNase-free tube that was instantly quick frozen with liquid nitrogen and stored at −80 °C for later analysis (transcriptome and gene expression). At the same time and with the same chickens, 60 samples of breast muscle (600 mg) were collected, placed in sterile tubes, and promptly snap-frozen in liquid nitrogen for metabolome analysis and stored at 80 °C until further processing. Meanwhile, enough samples were chopped and stored at −20 °C for fatty acid analysis.

### 2.2. Sample Preparation for RNA Sequencing Analysis

#### 2.2.1. RNA Extraction and Quality Assessment

The tissue of breast samples from five birds (male) at 100 days of age from each breed, which had a significant difference in meat quality and carcass traits, were selected. The total RNA was isolated using TaKaRa MiniREST Universal RNA Extraction Kit (TaKaRa Biotechnology Co., Ltd., Dalian, China), according to the manufacturer’s instructions. The purity and concentration of the RNA samples were evaluated by gel electrophoresis, A260/A280, and a NanoPhotometer1 spectrophotometer (Thermo Scientific, Wilmington, DE, USA). The integrity of the RNA samples was evaluated with the RNA Nano 6000 Assay Kit of the Bioanalyzer 2100 system (Agilent Technologies, California, CA, USA). Only samples with an RNA integrity number *>* 7 were used for constructing cDNA libraries and subsequent sequencing. The total RNA samples were kept at 80 °C until subsequent use.

#### 2.2.2. cDNA Library Preparation and Sequencing

RNA samples were selected for deep sequencing and library construction. Then, total RNA (7 µg) from each sample, ribosomal RNA was removed, and mRNA was enriched using oligo (dT) magnetic beads. cDNA libraries were synthesized using fragmented mRNAs that were generated as a template by using an RNA-seq sample preparation kit (Illumina, San Diego, CA, USA), following the manufacturer’s instructions. To complete library preparation, The cDNAs were afterward amplified using PCR; then, the cDNA library was paired-end sequenced with a 100-bp pair-end read length using an Illumina HiSeq 2500 platform [36]. 

### 2.3. Differential Expression Analysis

#### 2.3.1. Raw Data Quantification

The software of FastQC (Version 0.11.5, available online: https://github.com/s-andrews/FastQC accessed on 15 January 2022) was used to check raw sequence data. Clean sequence reads were obtained using the software Trim Galore (Version 0.4.4, available online: https://github.com/FelixKrueger/TrimGalore) accessed on 15 January 2022. Low-quality reads (more than half of the reads with a phred base quality score of less than 5) with adaptor and the unknown base were removed.

#### 2.3.2. Reads Alignment

The reads were then mapped to the Gallus gallus genome available online: (http://ftp.ensembl.org/pub/release-83/fasta/gallus_gallus/DNA/) accessed on 15 January 2022 using software HISAT2 (Version 2.0.4, available online: https://github.com/infphilo/hisat2 accessed on 15 January 2022) to obtain SAM files containing alignment accessed on 15 January 2022. Then, SAM files are converted to BAM files using Sam tools (Version 1.5, https://github.com/samtools/samtools for further analysis accessed on 15 January 2022.

#### 2.3.3. Gene Expression Analysis

The clean mapped reads were employed for gene expression analysis using DEseq2 available online: (https://github.com/mikelove/DESeq2) accessed on 15 January 2022. The padj (FDR) represented the significance of differentially expressed genes, and log2FoldChange (log2FC) represented relative gene expression level. Then, log2FC ≥ 1 and FDR < 0.05 were defined to obtain differentially expressed genes. The Benjamin–Hochberg false discovery rate was used to correct the *p*-values. Only genes with a *p*-value < 0.05 were considered differentially expressed [37].

### 2.4. Functional Enrichment Analysis

#### 2.4.1. Gene Ontology

The Gene Ontology database (available online: http://www.geneontology.org/) accessed on 15 January 2022 was performed to analyze pathways of differentially expressed genes and investigate the possible biological functions with the DAVID database (https://david.ncifcrf.gov/) accessed on 1 March 2022. GO pathways showing *p*-values of less than 0.05 were considered significantly enriched among the DEGs.

#### 2.4.2. Kyoto Encyclopedia of Genes and Genomes

The transcripts were analyzed on the KEGG online website (http://www.genome.jp/kegg/) accessed on 1 March 2022, and this database was used for investigating the possible biological functions. KEGG pathways showing *p*-values of less than 0.05 were considered significantly enriched among the DEGs [38].

#### 2.4.3. Protein–Protein Interaction (PPI)

The STRING 10 database was used to identify associations between the candidate genes identified in the present study (available online: http://string-db.org/.) accessed on 1 March 2022

### 2.5. Confirmation of the RNA-Seq Results via qRT-PCR

The reliability of the RNA-Seq analysis was confirmed using qRT-PCR analysis of 10 genes believed to affect the meat quality traits and muscle carcass traits of CH and DG chicken breeds. Primers were designed by Primer-BLAST on the NCBI website (https://www.ncbi.nlm.nih.gov/tools/primer-blast/) accessed on 1 March 2022 based on the Gallus_gallus-5.0 (NCBI) and synthesized by the Kunming synthesis department of TSINGKE biological technology Co., Ltd. (Kunming, China). Appendix A**,** for specific primer information. Seven samples were taken from each breed to collect tissue, the same chicken that was used for transcriptome analyses was used, and each sample was repeated three times. The total RNA of the breast muscle of chickens was then isolated and extracted using a kit produced by Bao Biological Engineering (Dalian) Co., Ltd., Dalian, China, according to the manufacturer’s instructions. cDNA was synthesized using the Prime Script RT reagent kit with gDNA Eraser (Takara, Dalian, China). qRT-PCR was performed using the Bio-Rad CFX96 real-time PCR platform (Bio-Rad Laboratories. lnc, California, CA, USA) with the SYBR^®^ Green PCR Master Mix Kit (Takara, Dalian, China). β-actin was employed as a reference gene for the normalization of gene expression levels. The fold change values were calculated using the 2-ΔΔCt method [39].

### 2.6. Statistical Analysis

Data were analyzed using the SPSS statistical software version 20.0 (IBM Corp., Armonk, New York, NY, USA). In this study, the means of the phenotypic indicators between CH and DG chicken breed were subjected to ANOVA testing. The means were assessed for significance by Tukey’s test, and t-tests were used to analyze the relevance between the two groups. The data were presented as Mean ± SEM, and *p* < 0.05 was used as a significant statistical difference.

## 3. Results

### 3.1. Phenotypic Characterization of Chicken

Performance traits of CH and DG chicken breeds at 100 days of age are shown in Table 1. The results showed that breed had a significant (*p* < 0.05) effect on body weight at 100 days of age. The BW of the DG chicken breed was significantly higher than that of the CH chicken breed (2646.4 vs. 1687.8, respectively). Table 1 shows the effect of genotype (CH and DG chicken breed) on breast weight parts at 100 days of age. The DG chicken breed had significantly (*p* < 0.05) higher breast muscle weight than the CH chicken breed.

There were no significant differences (*p* > 0.05) in the pH at 45 min after slaughtering between breast muscles of CH and DG chicken (6.2 vs. 6.3, respectively). However, the pH value of the breast at 24 h of CH chicken was significantly lower (5.6 vs. 5.9) than DG chicken (*p* < 0.05). The decline in the pH value of breast muscles (pH value at 45 min-pH value at 24 h) differed significantly (*p* < 0.05) between CH and DG breeds (0.52 vs. 0.33, respectively). The effect of genotype (CH and DG) on meat color measurements of breast muscles differed significantly between chicken breeds. The findings showed that L*, a* and b* values were 45.5, 6.9 and 8.2 for CH chickens, and 42.5, 3 and 3.4 for DG chickens (*p* < 0.05). Furthermore, the results revealed a significantly higher saturation index of breast (14.9) muscles in the CH breed compared to the DG breed with 6.4 (*p* < 0.05). WHC values in DG breast muscles were higher (62.2%) than in CH chicken (54.7) (*p* < 0.05).

EAAs: essential amino acid (including threonine, valine, methionine, isoleucine, leucine, phenylalanine, lysine, histidine, arginine, and proline); NEAAs: nonessential amino acids; TAAs: total amino acids.

The findings showed that the fatty acids following oleic acid (C18:1), linolenic acid (C18:3), and arachidonic acid (C20:4) content of breast meat was significantly higher in CH chicken than in DG chicken breed (11.2, 1.7, 18.9 vs. 3.8, 0.49, 9.38; *p* < 0.05), respectively. The findings also indicated that the content of total monounsaturated fatty acids (MUFA), total polyunsaturated fatty acids (PUFA), total unsaturated fatty acids (USFA), essential fatty acids (EFA), and total fatty acids (TFAC) in CH breast meat was significantly higher (*p* < 0.05) than that of DG meat with values of (17.1, 59.8, 76.9, 50.2, 107.4) and (8, 36.2, 44.2, 29.7), respectively. The results, as shown in Table 1 above, demonstrated that the amino acid composition of the two chicken meat types differed significantly. According to the results, threonine, serine, and phenylalanine amino acids were significantly greater (*p* < 0.05; 2.03, 1.5, 4.4 vs. 1.8, 1.3, 2.0), respectively, in the breast of the CH chicken breed compared with that of DG chicken. The results also showed that essential amino acids (EAAs), nonessential amino acid (NEAAs), and total amino acids (TAAs) content of CH breast meat were also significantly greater compared with that of DG meat (*p* < 0.05; 13.6, 34.8, 48.4 vs. 9.2, 31.8, 41.1, respectively).

### 3.2. Transcriptome Profiling of Chicken Breast Muscle

#### 3.2.1. Summary of Sequencing Data Statistics

The transcriptome data of 10 breast muscle tissues from five male Chahua chicken breeds and five Digao chicken breeds were employed in this study with the goal of understanding the molecular mechanism of chicken meat quality formation and carcass traits. We obtained more than 276 million clean reads ranging from 23.3 to 30 million reads, and 13,618 genes were detected, which corresponded to 11,016 to 11,764 expressed genes for each sample. The high-quality read of each sample was mapped to the jungle fowl reference genome Gallus_gallus-5.0; the average read mapping rate for each sample in CH and DG chicken was 83 and 85 %, respectively. The sequencing and mapped data statistics for each sample are presented in Table 2.

#### 3.2.2. Differentially Expressed Genes Analysis

To check the accuracy of biological samples, principal component analysis (PCA) was performed on the gene expression of 10 samples. The score plot of PCA analysis for two breeds shows that 55.4% of the variance can be explained by the first two principal components, accounting for 41.2 and 14% of the variance, respectively (Figure 1). Additionally, cluster analysis of DEGs of 10 tissues among CH and DG breeds was performed in each breed. Figure 2 depicts the clustering findings, which show that five chickens in each group were categorized together, significantly differentially expressed genes were significantly different, and there was good repeatability. It indicated that the breed factor was responsible for these differentially expressed genes.

Different columns represent different samples and different rows represent different genes. Varying colors represent the level of gene expression for the samples; the red color indicates high-level expression of genes, whereas the blue color means low level expression of genes.

In total 13,618 differentially expressed genes (DEGs) between CH and DG breeds were obtained by using the DESeq2 package. Many tests were then used, including Wald test, Bengamini, Hochberg, and padj (FDR), to correct these expressed genes, which selected only the significantly expressed genes with adjusted *p*-values of ≤0.05 and log2FC ≥ 0.1 FDR ≤ 0.05. Finally, we found 3525 DEGs of which 1750 were up-regulated and 1775 down-regulated in CH compared to DG. In this comparison, down-regulated expressed gene numbers were greater than up-regulated expressed gene numbers. Furthermore, the volcano plot of DEGs in CH vs. DG chicken breed Figure 3 showed significantly and not significantly expressed genes, with ten up-regulated and two down-regulated genes among the most significantly expressed genes. These DEGs could correlate to meat quality and carcass traits between CH and DG chicken breeds. The top 12 most significantly expressed genes in this comparison were *CHCHD10* (coiled-coil-helix-coiled-coil-helix domain containing 10), *MGARP* (mitochondria localized glutamic acid-rich protein), *COL28A1* (collagen type XXVIII α 1 chain), *CBARP* (CACN β subunit associated regulatory protein), ASB16 (ankyrin repeat and SOCS box containing 16), *SELENOW* (selenoprotein W), BORCS6 (BLOC-1 related complex subunit 6), PXDN (peroxidasin), TOM1 (target of myb1 membrane trafficking protein), TSC22D4 (TSC22 domain family member 4), SOD1 (superoxide dismutase 1), and *CRYAB* (crystallin α B). ASB16 (ankyrin repeat and SOCS box containing 16) gene expression in chicken breast muscle was first reported. The information on these genes is shown in Table 3.

### 3.3. Functional Analysis of Differentially Expressed Genes

#### 3.3.1. GO Enrichment Analysis

To explore the functions of the DEGs in chicken differentiation, DEGs were then used for GO analysis in this study to uncover their functional enrichment in this comparison. These DEGs were classified into three major GO categories of molecular function, cellular component, and biological process.

There were 2026 and 1928 enriched GO terms in GO enrichment down-regulated genes and GO enrichment up-regulated genes, respectively. However, there were 167 and 16 significantly enriched GO terms (*p <* 0.05) in down-regulated and up-regulated genes, respectively. These down-regulated genes were also divided into molecular function, cellular component, and biological process with enriched GO terms numbers 262, 234, and 1530, respectively; however, there were 20, 17, and 167 significantly enriched GO terms (*p <* 0.05), respectively. Furthermore, up-regulated genes were divided into molecular function, cellular component, and biological process with enriched GO terms numbers 267, 227, and 1433; however, in the CH vs. DG chicken breed comparison, there were 7, 10, and 12 significantly enriched GO terms (*p <* 0.05). Our outcomes showed that the DEGs in the CH vs. DG comparison were significantly (*p <* 0.5) enriched in GO terms down-regulated under the biological process that related to meat quality traits, such as regulation of phosphate metabolic process, axonogenesis and peptidyl-amino acid modification. Additionally, many pathways were found under the cellular component related to meat quality traits, such as collagen trimer, collagen-containing extracellular matrix, and cell junction actin cytoskeleton. as Additionally, there are many pathways found under molecular function that are related to meat quality, such as transmembrane receptor protein kinase activity, protein-containing complex binding, growth factor binding, collagen-binding protein tyrosine kinase activity and cytoskeletal protein binding, as shown in Table 4. The most significantly down-regulated enriched terms are shown in Appendix A.

Moreover, our results showed many of the significantly (*p <* 0.5) enriched GO upregulated biological process terms in the comparisons of CH and DG chicken are related to meat quality traits, such as the carboxylic acid metabolic process, oxoacid metabolic process, organic acid metabolic process, cellular amino acid metabolic process, cellular amino acid catabolic process and α-amino acid metabolic process. Additionally, there are pathways that significantly enriched under upregulated cellular components, such as mitochondrion, mitochondrial matrix and mitochondrial protein-containing complex. Furthermore, there are many pathways found under molecular function that are related to meat quality, such as heme binding, unfolded protein binding, iron ion binding and lyase activity, as shown in Table 4. The most significantly upregulated enriched terms are shown in Appendix A and Table 3.

#### 3.3.2. KEGG Enrichment Analysis

KEGG pathway analyses were also performed in this study to better understand DEG functions in chicken differentiation. KEGG analysis revealed that 147 and 140 enriched KEGG pathways were upregulated and downregulated, respectively. However, there were only 16 and 13 significantly downregulated and upregulated KEGG enriched pathways (*p <* 0.5), respectively, in this comparison. Many of these upregulated are significantly KEGG enriched. In the above-represented top are significantly enriched terms of upregulated genes. In the below represented top are significantly enriched terms of downregulated genes. 

Pathways are related to the meat quality, such as carbon metabolism, proteasome, valine, leucine, and isoleucine degradation, glyoxylate and dicarboxylate metabolism, fatty acid degradation, pyruvate metabolism and glycolysis/gluconeogenesis are shown in Table 5 In addition to KEGG down-regulated pathways, such as ECM-receptor interaction, the calcium signaling pathway, vascular smooth muscle contraction and melanogenesis are shown in Table 5. The most significantly upregulated and downregulated KEGG enriched terms are shown in Figure 4, Appendix A.

### 3.4. Construction of PPI Protein Interaction Network

The result of GO and KEGG showed that the significant and important DEGs that were involved or enriched in functions associated with meat quality traits that were shown in the tables above, such as GO terms of vascular smooth muscle contraction, melanogenesis, oxidative phosphorylation, carbon metabolism, valine, leucine and isoleucine degradation, fatty acid degradation, glyoxylate and dicarboxylate metabolism, pyruvate metabolism, tryptophan metabolism and glycolysis/gluconeogenesis pathways, are used in the STRING database. A PPI network, including a number of nodes: 78, number of edges: 184, average node degree: 4.72, avg. local clustering coefficient: 0.43, expected number of edges: 50, PPI enrichment *p*-value: <1.0 × 10^−16^ were built for these DEGs, as shown in Appendix A. The network has significantly more interactions than expected, and several DEGs played a core role in the PPI network.

### 3.5. Candidate Genes

Comprehensive analysis of DEGs, GO, KEGG and pathway results, and gene function enables us to recommend some genes such as *AGA, COL28A1, COL1A2, MB, HBAD, HBA1, ACACA, ACADL, ACSL1, ATP8A1, CAV1, FADS2, FASN, DCN, CHCHD10, AGXT2, ALDH3A2* and *MORN4* as the most promising candidate genes for having an impact on chicken meat quality. More details on these candidate genes above are listed in Table 6.

### 3.6. Validation Using qRT- PCR

A total of 10 genes have been selected for qRT-PCR confirmation of the RNA-seq data, including *COL28A1, COL1A2, MB, FASN, CHCHD10, IGF2, GHR, MYOD1, SMYD1* and *MYH10*. The figure shows that the relative expression trends between RNA-seq and qRT-PCR of the 10 genes (100 d DG chicken vs. 100 d CH chicken) were fully consistent, which supported the credibility of the RNA sequencing data. Any differences in relation to the ratios can be attributed to various algorithms and sensibilities of these technologies, as shown in Figure 5.

## 4. Discussion

Transcriptome sequencing via RNA resequencing or microarray technologies can be used to investigate the regulation of meat quality and carcass traits in chickens in a systematic manner, with some success in recent years.

The results found 3525 differentially expressed genes DEGs of which 1750 were up-regulated and 1775 down regulated in CH compared to DG chicken. Using RNA-seq to study the transcriptome of the breast muscle, [27] showed that 4608 differentially expressed genes (DEGs) were obtained by comparison of the Jinghai yellow. The CH chicken breed is a native breed distinguished by high-quality meat and slow growth. The DG chicken breed (broiler) is distinguished by low-quality meat and rapid growth. Therefore, evaluating the differences between these two breeds in regulatory pathways and genes is important to unravel the mechanisms of the DG breed’s rapid growth and the CH breed’s high-quality meat.

In this study, commercial chicken breed (DG) and native chicken (CH) breed have been used. These two breeds differ in meat quality traits and have a similar age. That’s why it is an ideal animal model to investigate the molecular mechanism of meat quality. In this study, RNA-Seq technique was used on breast muscle tissues from 100 days-old CH and DG broilers and identified a number of key DEGs that may affect meat quality.

There are many pathways and genes that are related to meat quality traits reported in this study. Sun et al. [5] revealed 14 genes as a candidate gene for meat quality, including *TYRO3* and *AGA* genes for intramuscular fat content in breast muscle, *COL28A1* and *COL1A2* for meat color and RET gene for abdominal fat traits when conducting a genome-wide association study to identify genes associated with the meat quality of Beijing-You chickens and Cobb-Vantress. breeds Li et al. [6] collected breast muscle samples of chicken and found several genes related to muscle color and amino acids metabolism, including *MB, HBAA, HBE1 HBM* and *HBE,* relating to the meat color and *PHGDH, DMGDH, AGXT* and *FTCD* are related to biosynthesis or metabolism of amino acids. Intriguingly, in our study, we also found many key genes that are linked to fatty acids synthesis, including DEGs (*ACADL; ACSL4; ECHS1; ECI1; ACAT2; HADH; HADHA; ECHS1; ACAA2; GCDH*) and analysis of KEGG results showed these genes were enriched in the fatty acid degradation pathway. Wang et al. [7] identified key genes that are associated with fatty acids synthesis, including *FASN, ACACA, HACD1, FADS2, ACSL1, HACD2,* and *ACSBG1* using transcriptome (RNA-Seq) in broiler chicken and KEGG analysis showed that genes were enriched in the pathway of fatty acid biosynthesis and biosynthesis of unsaturated fatty acids. Yang et al. [8] demonstrated promising candidate genes that regulate fatty acids composition by analyzing chicken transcriptomes from the thigh muscle tissue of Huangshan Black Chicken, including *DCN, FADS2, FRZB, OGN, CHCHD10, PRKAG3, ADGRD1* and *CYTL1*, are the most promising candidate genes affecting fatty acids composition using RNA sequencing technology. Remarkably *ACACA* can function as an important enzyme synthesized fatty acids. *ACACA* is an enzyme that can catalyze the synthesis of malonyl-CoA from two acetyl-CoA molecules and produce fatty acids under the action of fatty acid synthase [40]. Therefore, our study showed that the gene regulation associated with fatty acid metabolism was significantly greater in the CH chicken breed than the DG chicken breed, indicating the fat tissues in CH chicken have a stronger ability to synthesize the triglycerides cells. Thus, the expression of genes involved in fatty acid metabolism was significantly higher in CH chicken breeds compared to DG chicken breeds, and the content of fatty acids was also significantly higher in CH chicken breeds, implying that CH chicken breast muscles have a greater capacity for triglyceride synthesis. Moreover, our findings are consistent with the results of Zhang et al. [29], who investigated molecular mechanisms underlying meat quality and muscle carcass traits in the pectoral tissues of Gushi chickens and AA broilers.

The ultimate pH (pHu) trait is more important and more closely identified with chicken meat processing ability and sensory quality, and it is also influenced by the glycogen content of chicken muscle after slaughter. Beauclercq et al. [9] identified a biomarker that is related to meat pHu and glycogen metabolism by transcriptome analysis using microarray technology in the breast muscle of two chicken lines differentiated by pHu values, including *MORN4* and *MYLIP* genes.

Meat color is an essential factor that may influence customers’ buying decisions [41]. Many studies have reported that color of breast muscle was higher in several indigenous chicken breeds [42,43]. These are all changeable in parameters of meat color, maybe back to diluting of haem pigments. Our results are similar to Li et al. [6] who studied the molecular mechanisms of chicken meat color and taste using RNA-seq-based quantitative transcriptome analysis. The top 10 enriched pathways in his study included glycolysis/gluconeogenesis, carbon metabolism, heme binding, hemoglobin complex and haemoglobin binding. Numbers of DEGs were also identified, such as hemoglobin subunit epsilon (*HBE*), hemoglobin subunit epsilon 1 (*HBE1*), and *MB*, which were upregulated genes related to red meat color. Moreover, KEGG enrichment analysis identified DEGs related to amino acid metabolism all of which are connected with metabolism or amino acid biosynthesis, including *CHDH* and *AGXT*. The variable in meat color is believed to be caused by a complex interaction between the met-MB control enzyme system and oxidative processes [44]. The color of meat is determined by the amount of oxy-MB and MB in the muscle, which determines the color, such as red, bright, and pink that consumers may desire. This relationship between oxy-MB and MB content has been investigated [45,46].

MB has a key role in color and promotes more than 80% of the pigment of meat, and the *MB* gene expression is key to regulating meat color [47,48]. The current study suggested that the *MB* gene was upregulated and was able to function in oxygen binding and heme-binding, and it may also be related to redness color value. An additional study reported that *MB* gene expression was correlated to a∗ values in pork meat [49]; hemoglobin is a major factor in the formation of red meats color [50].

Therefore, *MB, HBAD,* and *HBA1* genes were assumed as key candidate genes that can play an important role in promoting meat color in chickens; they encode subunits of hemoglobin and all of them were upregulated RT-qPCR and RNA-seq results.

## 5. Conclusions

In conclusion, this study has created the transcriptome profiles of the breast muscle from two chicken breeds (DG c and CH), which have different meat quality at the age of 100 days, using RNA sequencing technology. Successive bioinformatic analyses indicated that some DEGs, such as *AGA, COL28A1, COL1A2, MB, HBAD, HBA1, ACACA, ACADL, ACSL1, ATP8A1, CAV1, FADS2, FASN, DCN, CHCHD10, AGXT2, ALDH3A2* and *MORN4* and pathways, such as oxidative phosphorylation, carbon metabolism, valine, leucine and isoleucine degradation, fatty acid degradation, glyoxylate and dicarboxylate metabolism, pyruvate metabolism, tryptophan metabolism and glycolysis/gluconeogenesis, might be indispensable for the regulation of meat quality traits. This comparative transcriptome analysis of Digao and Chahua muscle has been reported for the first time. This study is useful in forecasting new gene functions and in the exploration of the mechanism of meat quality traits process in chickens.

## Figures and Tables

**Figure 1 genes-14-00095-f001:**
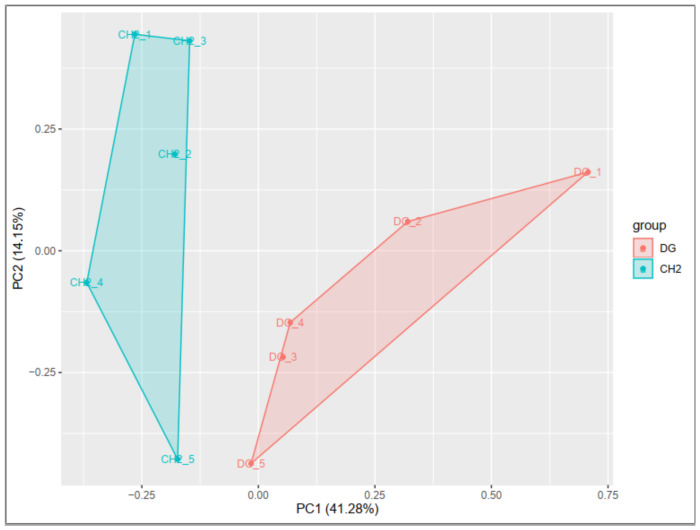
PCA of the expression levels of 10 samples across CH and DG chicken breeds.

**Figure 2 genes-14-00095-f002:**
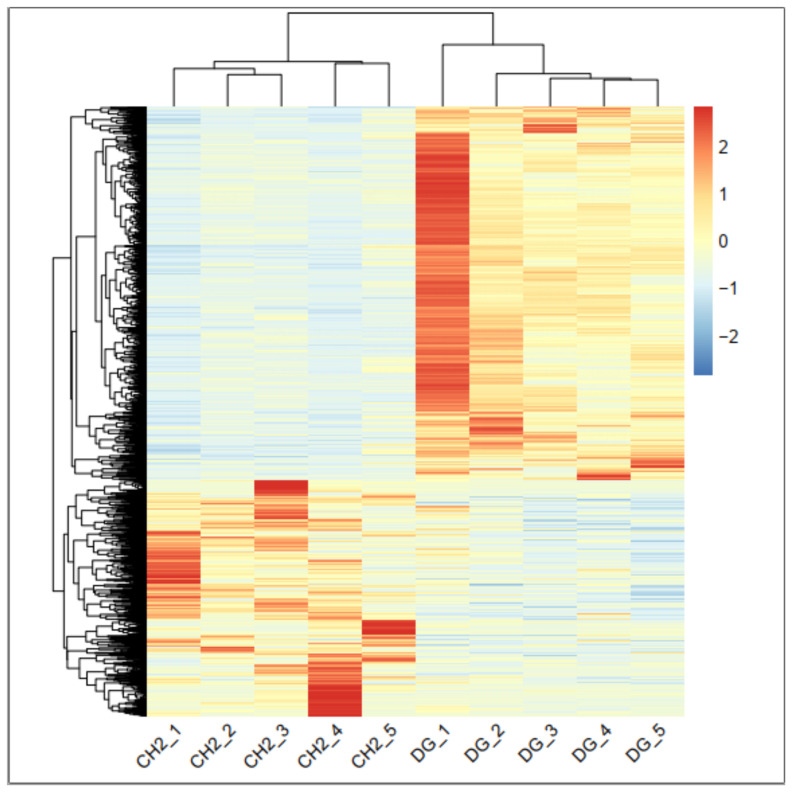
The cluster analysis for DEGs of CH vs. DG.

**Figure 3 genes-14-00095-f003:**
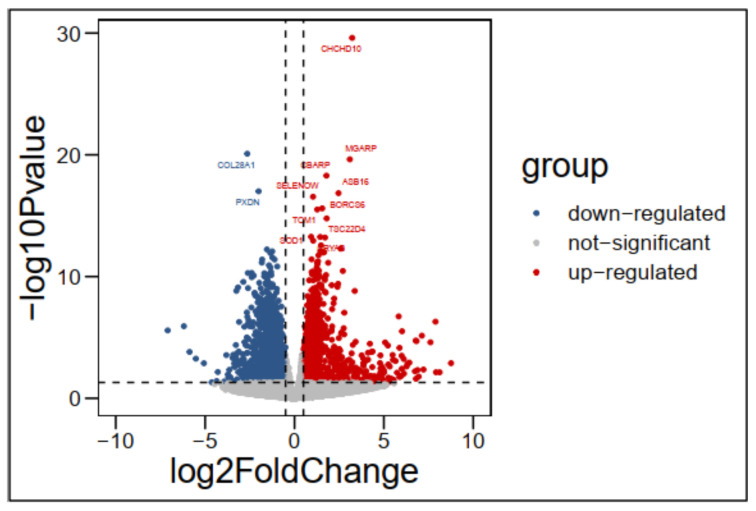
The volcano plot of DEGs in CH vs. DG.

**Figure 4 genes-14-00095-f004:**
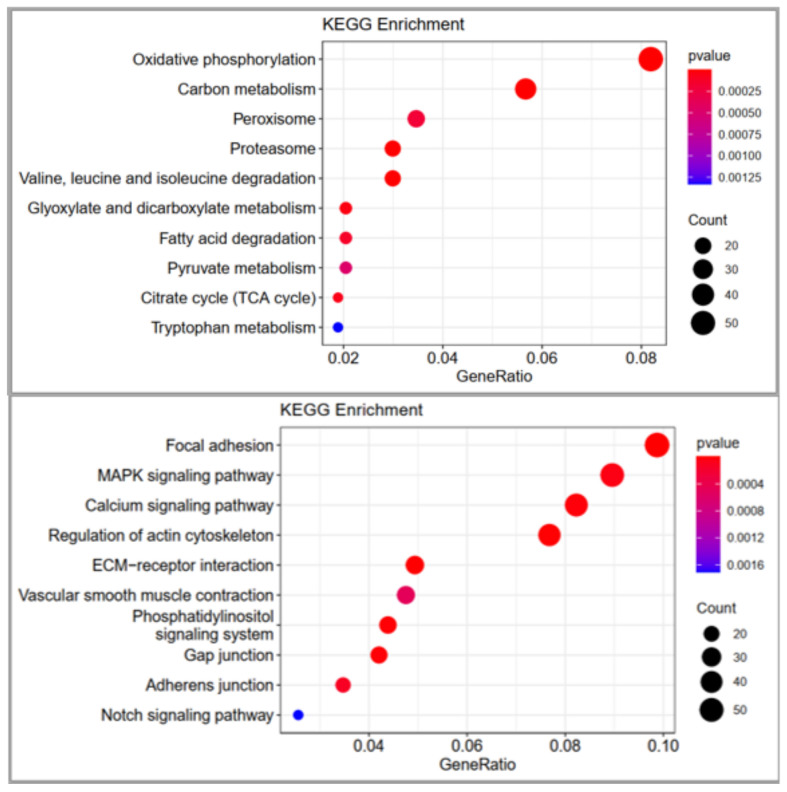
The most significantly Kyoto Encyclopedia of Genes and Genomes pathway analysis of differentially expressed genes between CH and DG chicken.

**Figure 5 genes-14-00095-f005:**
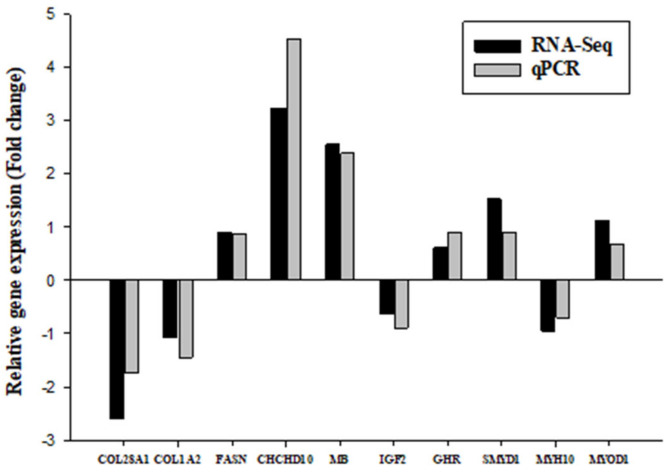
qRT-PCR validation of the gene expression profiles. The relative expression trends between RNA-seq and qRT-PCR of the 10 genes (100 d DG chicken vs. 100 d CH chicken) were completely consistent. The fold change represents the ratio of average expression of native samples relative to that of commercial breed.

**Table 1 genes-14-00095-t001:** Performance traits of breast muscle of CH and DG chicken breeds.

Meat Quality Parameters	Genotype		*p*- Value
	CH	DG	
** *Physical traits* **			
pH 45 min	6.2a ± 0.06	6.3a ± 0.03	0.1729
pH 24 h	5.6b ± 0.05	5.9a ± 0.02	<0.0001
pH change	0.52a ± 0.05	0.33b ± 0.02	0.0020
lightness *(L*)*	45.5a ± 1	42.5b ± 0.5	0.0100
Redness *(a*)*	6.9a ± 0.3	3.0b ± 0.1	<0.0001
Yellowness *(b*)*	8.2a ± 0.4	3.4b ± 0.2	<0.0001
Saturation index	14.9a ± 0.6	6.4b ± 0.3	<0.0001
(WHC) %	54.7b ± 2	62.2a ± 1	0.0020
** *Fatty acids composition (mg\g)* **			
C18:1	11.2a ± 1.5	3.8b ± 1.5	0.0122
C18:3, cis-9,12,15	1.7a ± 0.3	0.49b ± 0.3	0.0246
C20:4	18.9a ± 1.8	9.38b ± 1.8	0.0093
PUFA	59.8a ± 4.6	36.2b ± 4.6	0.0092
USFA	76.9a ± 6.6	44.2b ± 6.6	0.0106
EFA	50.2a ± 3.9	29.7b ± 3.9	0.0086
TFAC	107.4a ± 9.2	69.3b ± 9.2	0.0232
** *Amino acid profile (mg\g)* **			
Aspartic acid	2.2a ± 0.04	2.1b ± 0.04	0.054
Threonine	2.03a ± 0.06	1.8b ± 0.06	0.0361
Serine	1.5a ± 0.04	1.3b ± 0.04	0.0237
Glutamic acid	4.7a ± 0.4	3.2b ± 0.4	0.0595
Citruline	4.4a ± 0.07	4.2b ± 0.07	0.0556
Phenylalanine	4.4a ± 0.6	2.0b ± 0.6	0.0299
EAAs	13.6a ± 1	9.2b ± 1	0.0256
NEAAs	34.8a ± 0.8	31.8a ± 0.8	0.047
TAAs	48.4a ± 1.1	41.1a ± 1.1	0.0026

Means with the different letter in the same row are significantly different (*p* < 0.05) among Chahua chicken breed and Digao chicken breed or male and female. Data are means ± SD; USFA: unsaturated fatty acids; PUFA: polyunsaturated fatty acids; TFAC: total fatty acids; EFA: essential fatty acids (including linoleic acid, linolenic acid, and arachidonic acid).

**Table 2 genes-14-00095-t002:** Sequencing and mapped data statistics.

Sample	Reads	Genome Mapping Rate (%)	Gene Mapping Rate (%)	Expressed Genes
** *CH2_1* **	26,023,785	83.55%	80.89%	11,016
** *CH2_2* **	27,365,683	82.20%	82.66%	11,257
** *CH2_3* **	27,896,323	83.56%	83.69%	11,397
** *CH2_4* **	23,3995,48	80.71%	85.94%	11,703
** *CH2_5* **	24,661,032	85.37%	83.99%	11,438
** *DG_1* **	30,316,157	85.04%	86.39%	11,764
** *DG_2* **	30,788,217	85.53%	84.16%	11,461
** *DG_3* **	26,681,249	83.81%	84.69%	11,533
** *DG_4* **	30,609,060	85.46%	83.93%	11,430
** *DG_5* **	28,357,362	85.80%	84.40%	11,493

CH2: Chahua chicken breed, DG: Digao chicken breed. The mapping ratio indicates the level of total mapped reads as a proportion of clean reads.

**Table 3 genes-14-00095-t003:** Expression levels of differentially expressed genes with the most significant differences.

Gene	Base Mean	log2 Fold Change	lfcSE	Stat	*p*-Value	padj	Up/Down CH vs. DG
** *CHCHD10* **	10448.7	3.230314	0.282229	11.44574	2.47 × 10^−30^	2.75 × 10^−26^	UP
** *MGARP* **	256.572	3.09816	0.3351	9.245478	2.34 × 10^−20^	8.70 × 10^−17^	UP
** *COL28A1* **	1739.54	−2.64248	0.282382	−9.35781	8.14 × 10^−21^	4.54 × 10^−17^	DOWN
** *CBARP* **	761.393	1.788038	0.200703	8.908884	5.15 × 10^−19^	1.44 × 10^−15^	UP
** *ASB16* **	3192.03	2.462107	0.288495	8.534301	1.41 × 10^−17^	2.62 × 10^−14^	UP
** *SELENOW* **	21137.9	1.038493	0.122805	8.456458	2.76 × 10^−17^	4.39 × 10^−14^	UP
** *BORCS6* **	1907.88	1.551992	0.189377	8.19523	2.50 × 10^−16^	3.48 × 10^−13^	UP
** *PXDN* **	647.359	−2.00603	0.233939	−8.57502	9.91 × 10^−18^	2.21 × 10^−14^	DOWN
** *TOM1* **	25202.6	1.272602	0.155792	8.16858	3.12 × 10^−16^	3.86 × 10^−13^	UP
** *TSC22D4* **	10700.1	1.802172	0.226302	7.963585	1.67 × 10^−15^	1.86 × 10^−12^	UP
** *SOD1* **	7424.02	0.929765	0.123612	7.521614	5.41 × 10^−14^	5.20 × 10^−11^	UP
** *CRYAB* **	7348.54	1.440085	0.191574	7.517112	5.60 × 10^−14^	5.20 × 10^−11^	UP

**Table 4 genes-14-00095-t004:** Significantly enriched terms of GO pathways related to the meat quality.

ID	Description	*p*. Adjust	Meat Quality Related Genes	DEGs Number
** *Down regulated pathways* **				
** *GO:0009893* **	positive regulation of metabolic process	0.01471	*ALDH1A2; ACACA; NFIB; TADA2A; CDH13; ACTA2; DCN*	71
** *GO:0043062* **	extracellular structure organization	2.60 × 10^−5^	*COL6A1; COL1A2; COL3A1; COL8A1*	18
** *GO:0005581* **	collagen trimer	0.0017	*COL6A3; COL6A1; COL1A2; COLEC12; COL8A1; COLEC10; COL3A1; OL14A1; COL6A2; COLEC11*	11
** *Upregulated pathways* **				
** *GO:0006520* **	cellular amino acid metabolic process	0.00488	*GOT2; YARS; HMGCL; BLMH; FTCD; GOT1; TST; GLDC; ADI1; AGMAT; FARSA; ASNSD1; HIBCH; GLUL*	14
** *GO:0009063* **	cellular amino acid catabolic process	0.00039	*GOT2; HMGCL; BLMH; FTCD; GLDC; HIBCH*	6
** *GO:0020037* **	heme binding	0.00053	*CYP2C23b; SDHD; CYCS; MB; CYB5A; HBAD; SUOX; SLC48A1; HBA1; CYP2C23a*	10

**Table 5 genes-14-00095-t005:** The significantly enriched terms of KEGG pathways related to the meat quality traits.

ID	Description	*p*. Adjust	Meat Quality-Related Genes	DEGs Number
**Down regulated pathways**				
** *gga04270* **	Vascular smooth muscle contraction	0.0075	*CALCRL; KCNMA1; ADORA2A; MYH11; CACNA1C*	26
** *gga04916* **	Melanogenesis	0.0345	*CAMK2A; GNAI1; CTNNB1; KIT*	19
** *Upregulated pathways* **				
** *gga00190* **	Oxidative phosphorylation	7.66 × 10^−16^	*UQCR11; COX10; NDUFB6; ATP6V0E1*	52
** *gga01200* **	Carbon metabolism	6.26 × 10^−8^	*KCNMA1; GOT2; PGAM1; GLDC; RGN; PGK2*	36
** *gga00280* **	Valine, leucine and isoleucine degradation	1.49 × 10^−5^	*AGXT2; HMGCL; HADH; ECHS1; HADHA*	
** *gga00071* **	Fatty acid degradation	0.0027	*ACADL; ACSL4; ECHS1; ECI1; ACAT2; HADH; HADHA; ECHS1; ACAA2; GCDH*	13
** *gga00630* **	Glyoxylate and dicarboxylate metabolism	0.00123	*SHMT1; GLDC; GLUL; GLYCTK*	13
** *gga00620* **	Pyruvate metabolism	0.0077	*PC; PKLR; ACYP1; ACAT2*	13
** *gga00380* **	Tryptophan metabolism	0.0195	*ALDH3A2; HAAO; AOX2; ECHS1*	12
** *gga00010* **	Glycolysis / Gluconeogenesis	0.0445	*ALDH3A2; DLD; ALDOB; PDHA2*	14

**Table 6 genes-14-00095-t006:** Detailed information about the candidate genes that related to meat quality.

Trait	Gene	log2 Fold Change	*p*-Value	*p*-Adjust	Up/Down
** *Meat color* **	*COL28A1*	−2.6	8.14 × 10^−21^	4.54 × 10^−17^	DOWN
	*COL1A2*	−1.08	0.006	0.028	DOWN
	*MB*	2.55	0.0005	0.0042	UP
	*HBAD*	2.44	0.0020	0.011	UP
	*HBA1*	2.38	0.00049	0.0040	UP
** *Fatty acid* **	*ACACA*	−0.70	0.00035	0.0031	DOWN
	*ACADL*	0.949	0.0003	0.0031	UP
	*ACSL1*	0.559	6.96 × 10^6^	2.16 × 10^−5^	UP
	*ATP8A1*	−0.786	0.0030	0.015	DOWN
	*CAV1*	0.168	2.05 × 10^−9^	2.38 × 10^−7^	UP
	*FADS2*	−3.59	0.00032	0.0029	DOWN
	*FASN*	0.90	4.61 × 10^−6^	0.00011	DOWN
	*DCN*	−0.845	0.0068	0.029	DOWN
	*CHCHD10*	3.23	2.47 × 10^−30^	2.75 × 10^−26^	UP
** *Amino acids* **	*AGXT2*	2.79	9.29 × 10^−8^	4.88 × 10^−6^	UP
	*ALDH3A2*	0.754	9.39 × 10^−7^	3.20 × 10^−5^	UP
** *pH* **	*MORN4*	1.09	0.0001	0.001	UP

## Data Availability

RNA resequencing data fastq files will be available in the NCBI Sequence Read Archive under Bio Project accession number PRJNA876777 and some will be available on request from the corresponding author. The data are not publicly available due to privacy.

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
