# Peer review of "Integrated Transcriptomics Profiling in Chahua and Digao Chickens’ Breast for Assessment Molecular Mechanism of Meat Quality Traits"

_genes, 2022, doi:10.3390/genes14010095_

Round 1

Reviewer 1 Report

Dear authors,

Your manuscript entitled - Integrated Transcriptomics Profiling in Chahua and Digao Chickens’ Breast for Assessment Molecular Mechanism of Meat Quality Traits- is  written very clearly and the paper will be an excelent reserach addition to the existing literature and research field.

I appreciate your work, you succeded to explain your reserch in a very simple and clear writing, with an intersting content .

Abstract

A very comprehensive abstract, using concentrated information of  manuscript.

Introduction

I find the introduction chapter too long. Usually the introduction is the shortest chapter of a manuscript. Please, short the length of the introduction.

Row 70 – Over....small letters

Row 104, 106 – Please mention the authors....not only the parenthesis

Row- 116 Is it I2? Or is it 12 weekes of age? Please corect.

Materials and Methods

Row 161, 164- use space tab between number and unit of measure 200 mg, 600 mg

Summary of sequencing data statistics

Row 283 - Please verify the font style within text

3.3.2. KEGG enrichment analysis

Row 375 – Start the sentence from the beginning

Row 378 – Capital letter on pathways

Row 388-396 - Please verify the font style within text

Table 3 Please use BOLD for Downregulated pathways and decide if is one word written or two words as in Table 4?

Table 4 – Please decide if downregulated  pathways is centered as upregulated pathways or not??

Discussion

Row 428-504 - Please verify the font style within text

Row 434, 443, 461- Please mention the authors....not only the parenthesis, at the beginning of a sentence/phrase

Conclusion

Row 495 – you mentioned chicken breeds, so use in parenthesis just (DG and CH)

Row 499 – small letters for  Oxidative, Carbon, Valine

Author Response

We are very grateful for the reviews provided of this manuscript and we appreciate your comments and suggestions. The comments are encouraging, and the reviewers appear to share our judgement that this study and its results are clinically important.

Our statistician made the following comments:
E.1. Reviewer comment: Abstract: A very comprehensive abstract, using concentrated information of manuscript.

Authors’ Response: Thank you for your encouragement

E.2. Reviewer comment: I find the introduction chapter too long. Usually the introduction is the shortest chapter of a manuscript. Please, short the length of the introduction.

Authors’ Response: We thank the Reviewer for having suggested this important point, but We have tried to cover all aspects related to this study in order to give the reader a deep understanding about the study.

E.3. Reviewer comment: Row 70 – Over....small letters.

Authors’ Response: This point was revised.

E.4. Reviewer comment: Row 104, 106 – Please mention the authors....not only the parenthesis.

Authors’ Response: This point was revised.

E.5. Reviewer comment: Row- 116 Is it I2? Or is it 12 weeks of age? Please correct.

Authors’ Response: This point was corrected.

E.6. Reviewer comment: ow 161, 164- use space tab between number and unit of measure 200 mg, 600 mg.

Authors’ Response: This point was corrected.

E.7. Reviewer comment: Row 283 - Please verify the font style within text

Authors’ Response: This point was corrected.

E.8. Reviewer comment: Row 375 – Start the sentence from the beginning

Authors’ Response: This sentence is describing the figure.

E9. Reviewer comment: Row 378 – Capital letter on pathways

Authors’ Response: This sentence is continuing to sentence before the figure.

E10. Reviewer comment: Row 388-396 - Please verify the font style within text

Authors’ Response: This point was revised

E11. Reviewer comment: Table 3 Please use BOLD for Downregulated pathways and decide if is one word written or two words as in Table 4?

Authors’ Response: This point was revised

E12. Reviewer comment: Table 4 – Please decide if downregulated  pathways is centered as upregulated pathways or not??

Authors’ Response: This point was revised

E13. Reviewer comment: Row 428-504 - Please verify the font style within text

Authors’ Response: This point was revised

E14. Reviewer comment: ow 434, 443, 461- Please mention the authors....not only the parenthesis, at the beginning of a sentence/phrase

Authors’ Response: This point was revised

E15. Reviewer comment: Row 495 – you mentioned chicken breeds, so use in parenthesis just (DG and CH)

Authors’ Response: This point was revised

E16. Reviewer comment: Row 499 – small letters for  Oxidative, Carbon, Valine

Authors’ Response: This point was revised

First, we appreciate your detailed and kind considerations at our manuscript. We understand your concern about the attention to detail and inclusion of additional studies. We have done our best to revise the mistakes and add the information suggested.

Reviewer 2 Report

1) The purpose of the study was not clear in the abstract, please describe.

2) Lines 170-178: Why did you use only 5 samples (birds) from each strain for this analysis? The sample number is not enough to demonstrate a reliable result at the statistical level.

3) I think there could be a topic in the material and methods part about "Statistical Analysis" giving a complete description of how it was performed.

4) I could not access the supplementary tables that are the meat quality data. I find it interesting that you add a table with all the quality analyzes performed, in the article itself, even for the reader, I was able to relate it in a simpler way, analyzing the meat quality results with the gene expression that you made, differentiating the two strains.

Author Response

We are very grateful for the reviews provided of this manuscript and we appreciate your comments and suggestions. The comments are encouraging, and the reviewers appear to share our judgement that this study and its results are clinically important.

E1. Reviewer comment: The purpose of the study was not clear in the abstract, please describe.

Authors’ Response: This point was changed, Meat quality traits are an important economic trait and it remain a major argument, from the producer to the consumer. T However, there are a few candidate genes and pathways of chicken meat quality traits that were reported for chicken molecular breeding. The purpose of the present study is to identify the candidate genes and pathways associated with meat quality underlying variations in meat quality. Hence, transcriptome profiles of breast tissue in commercial DG (5 male) and CH (5 male) native chicken breeds at the age of 100 days were analyzed.

E2. Reviewer comment: Lines 170-178: Why did you use only 5 samples (birds) from each strain for this analysis? The sample number is not enough to demonstrate a reliable result at the statistical level.

Authors’ Response: I have read more than 50 articles, and most of these studies have used sample numbers ranged from 3 to 6 samples for RNA seq and biological replicates are required if inference on the population is to be made, with three biological replicates being the minimum for any inferential analysis.  In our study we have used 5 samples and its enough to demonstrate a reliable result at the statistical level. In this study we want a list of the most highly DE genes from samples that are fairly stereotypical, five full biological replicates suffice. However, I agree with you to increase the number of replicates and samples. As a general rule, the number of biological replicates should never be below 3. For a basic RNA-seq differential expression experiment, 10M to 20M reads per sample is usually enough. If similar data exists it can be helpful to check the read counts for key genes of interest to estimate the required depth. The number of replicates you decide to sequence is quite literally a cost-benefit analysis since the major limiter these days seems to be the price of sequencing.

 E3. Reviewer comment: I think there could be a topic in the material and methods part about "Statistical Analysis" giving a complete description of how it was performed

Authors’ Response: we have added a topic in the material and methods part about "Statistical Analysis “and the statistical analysis for RNA seq already confirmed in advance

Statistical analysis
Data were analyzed using the SPSS statistical software version 20.0 (IBM Corp., Armonk, New York). In this study, the means of the phenotypic indicators between CH and DG chicken breed were subjected to ANOVA testing. The means were assessed for significance by Tukey’s test, and t-tests were used to analyze the relevance between the two groups. The data were presented as Mean ± SEM, and P < 0.05 was used as a significant statistical difference.

E4. Reviewer comment: I could not access the supplementary tables that are the meat quality data. I find it interesting that you add a table with all the quality analyzes performed, in the article itself, even for the reader, I was able to relate it in a simpler way, analyzing the meat quality results with the gene expression that you made, differentiating the two strains.

Authors’ Response: we have added a table with all the quality analyzes performed

Table 1. Performance traits of breast muscle of CH and DG chicken breeds

Meat quality parameters

Genotype

P- value

CH

DG

Physical traits

pH 45 min

6.2a±0.06

6.3a±0.03

0.1729

pH 24 h

5.6b±0.05

5.9a±0.02

<.0001

pH change

0.52a±0.05

0.33b±0.02

0.0020

L*

45.5a±1

42.5b±0.5

0.0100

a*

6.9a±0.3

3.0b±0.1

<.0001

b*

8.2a±0.4

3.4b±0.2

<.0001

Saturation index

14.9a±0.6

6.4b±0.3

<.0001

 (WHC) %

54.7b±2

62.2a±1

0.0020

fatty acids composition (mg\g)

C18:1

11.2a±1.5

3.8b±1.5

0.0122

C18:3, cis-9,12,15

1.7a±0.3

0.49b±0.3

0.0246

C20:4

18.9a±1.8

9.38b±1.8

0.0093

PUFA

59.8a±4.6

36.2b±4.6

0.0092

USFA

76.9a±6.6

44.2b±6.6

0.0106

EFA

50.2a±3.9

29.7b±3.9

0.0086

TFAC

107.4a±9.2

69.3b±9.2

0.0232

Amino acid profile (mg\g)

Aspartic acid

2.2a±0.04

2.1b±0.04

0.054

Threonine

2.03a±0.06

1.8b±0.06

0.0361

Serine

1.5a±0.04

1.3b±0.04

0.0237

Glutamic acid

4.7a±0.4

3.2b±0.4

0.0595

Citruline

4.4a±0.07

4.2b±0.07

0.0556

Phenylalanine

4.4a±0.6

2.0b±0.6

0.0299

EAAs

13.6a±1

9.2b±1

0.0256

NEAAs

34.8a±0.8

31.8a±0.8

0.047

TAAs

48.4a±1.1

41.1a±1.1

0.0026

Means with the different letter in the same row are significantly different (P<0.05) among Chahua chicken breed and Digao chicken breed or male and female.

Data are means ± SD; USFA: unsaturated fatty acids; PUFA: polyunsaturated fatty acids; TFAC: total fatty acids; EFA: essential fatty acids (including linoleic acid, linolenic acid, and arachidonic acid).

EAAs: essential amino acid (including threonine, valine, methionine, isoleucine, leucine, phenylalanine, lysine, histidine, arginine, and proline); NEAAs: nonessential amino acids; TAAs: total amino acids.

Reviewer 3 Report

In the manuscript "Integrated Transcriptomics Profiling in Chahua and Digao Chickens' Breast for Assessment Molecular Mechanism of Meat Quality Traits" The authors presented interesting data concerning the molecular mechanism of meat quality traits. Overall, the manuscript is well-written but it still needs to be grammar, format, and punctuation checks. I have a few comments, which can enrich this manuscript to be more interesting to readers.

1.     Line 21 (Abstract) When DG and CH appear for the first time, the full name should be used.

2.     Line 23 (Abstract) The abbreviations (DEGs) should be taken into brackets.

3.     Line 59 All gene names must be in italic.

4.     Line 170 (Materials and Methods) What was the rationale for choosing the 100 days of age?

5.     Line 187 (Materials and Methods) I noticed the cDNA library was paired-end sequenced with a 100-bp pair-end read length using an Illumina HiSeq 2500 platform. Whether the sequencing was completed in your laboratory? In addition, the RIN parameter informs about the RNA integrity is important for NGS sequencing, please provided each sample RIN information.

6.     Line 193 (Materials and Methods) What do you define as “Low-quality reads”, Please list the detailed parameters.

7.     Line 205 (Materials and Methods) Except for genes with Log2FC > 1, how the Log2FC < -1 genes were considered?

8.     Line 399 (Results) How many genes were employed to generate the PCA plots?

9.     Line 300 (Results) How many genes were employed to generate the heatmap? Please indicate the number of genes used and the reason for selection.

10.  Line 311 (Results) Figure3.3 should be Figure 3.

11.  Line 324-327 (Results) This part of content is redundant, please deleted it.

12.  Since the objective of this study was to assess the molecular mechanism of meat quality traits, can the authors propose a mechanism regulate module based on the DEGs identified in this study?

Author Response

In the manuscript "Integrated Transcriptomics Profiling in Chahua and Digao Chickens' Breast for Assessment Molecular Mechanism of Meat Quality Traits" The authors presented interesting data concerning the molecular mechanism of meat quality traits. Overall, the manuscript is well-written, but it still needs to be grammar, format, and punctuation checks. I have a few comments, which can enrich this manuscript to be more interesting to readers.

Authors’ Response: Answer: Thank you for reviewing our manuscript. The manuscript was revised by an expertise in English language, following the suggestions. The grammar errors were corrected appropriately.

E1. Reviewer comment: Line 21 (Abstract) When DG and CH appear for the first time, the full name should be used.

Authors’ Response: we have corrected this point.

E2. Reviewer comment: Line 23 (Abstract) The abbreviations (DEGs) should be taken into brackets

Authors’ Response: we have corrected this point.

E3. Reviewer comment: Line 59 All gene names must be in italic

Authors’ Response: we have corrected this point.

E4. Reviewer comment: Line 170 (Materials and Methods) What was the rationale for choosing the 100 days of age?

Authors’ Response: This chicken aged was appropriate for this kind of chicken breeds because CH chickens are commonly slaughtered on the I2 to 16 weeks (90 to 120day) of age and DG chickens are slaughtered on the 8 to 16 weeks. In our study we tried to choose that aged led to collecting the sample at same aged and within marketed period

E5. Reviewer comment: Line 187 (Materials and Methods) I noticed the cDNA library was paired-end sequenced with a 100-bp pair-end read length using an Illumina HiSeq 2500 platform. Whether the sequencing was completed in your laboratory? In addition, the RIN parameter informs about the RNA integrity is important for NGS sequencing, please provided each sample RIN information.

Authors’ Response: we have added this point and the sequencing was completed in the company of Biolinker Technology (Kunming) CO., Ltd. your laboratory and only samples with an RNA integrity number > 7 were used for constructing cDNA libraries and subsequent sequencing.

E6. Reviewer comment: Line 193 (Materials and Methods) What do you define as “Low-quality reads”, Please list the detailed parameters.

Authors’ Response: we have added this point and “Low-quality reads (more than half of the reads with a phred base quality score of less than 5).

E7. Reviewer comment:  Line 205 (Materials and Methods) Except for genes with Log2FC > 1, how the Log2FC < -1 genes were considered?

Authors’ Response: Generally, log2FC< -1, log2FC>1 and P<0.01 are selected.

E8. Reviewer comment:  Line 399 (Results) How many genes were employed to generate the PCA plots?

Authors’ Response:  This point already minted in Summary of sequencing data statistics paragraph

E9. Reviewer comment:  Line 300 (Results) How many genes were employed to generate the heatmap? Please indicate the number of genes used and the reason for selection

Authors’ Response:  This point already minted in Summary of sequencing data statistics paragraph and usually the number of differential genes selected from the above conditions is in the range of several hundreds to several thousands. So, we adjusted the threshold value according to our expected results.

E10. Reviewer comment: Line 311 (Results) Figure3.3 should be Figure 3.

Authors’ Response:  we have corrected this point.

E11. Reviewer comment: Line 324-327 (Results) This part of content is redundant, please deleted it.

Authors’ Response:  we have deleted this point.

E12. Reviewer comment: 12.  Since the objective of this study was to assess the molecular mechanism of meat quality traits, can the authors propose a mechanism regulate module based on the DEGs identified in this study?

Authors’ Response:  Thank you so much for this important comment, in actually I want to inform you that there are many traits and parameters for describing meat quality and it’s difficult to propose a mechanism regulate module that can linked all traits and genes obtained, so I agree with you to change molecular mechanism to identifying DEGs and pathway associated with meat quality.

Round 2

Reviewer 2 Report

The authors carried out all the suggestions given by the reviewer, except one, explaining in a succinct and technical way why not. I consider the article ready for publication.